# Research on a Charging Mechanism of Electric Vehicles for Photovoltaic Nearby Consumption Strategy

**Qingsu He** [1,2,*] **, Muqing Wu** [1,*] **, Pei Sun** [2] **, Jinglin Guo** [3] **, Lina Chen** [2] **, Lihua Jiang** [3] **and Zhiwei Zhang** [1]

1    Beijing Laboratory of Advanced Information Networks, Beijing Key Laboratory of Network System Architecture and Convergence, Beijing University of Posts and Telecommunications, Beijing 100876, China
2    State Grid Gansu Electric Power Company, Lanzhou 730050, China
3    State Grid Information and Communication Industry Group Corporation, Beijing 100010, China
*    Correspondence: heqingsu@bupt.edu.cn (Q.H.); wumuqing@bupt.edu.cn (M.W.)

**Abstract:** With the promotion of the pilot development of distributed whole county roof photovoltaics in China, problems such as power consumption, energy regional balance, and grid stability have become prominent. In this paper, an application mode of electric vehicle (EV) charging network and distributed photovoltaic power generation local consumption is studied. The management idea of two-layer and four model has been established, including the regional distributed photovoltaic output model, electricity consumption model, EV consumption model, and regional grid load dispatching model, which can realize the scheduling of the energy flow formed by photovoltaic, induce the charging of EVs, and make the photovoltaic consumption in office building areas and residential building areas complementary. Firstly, according to the randomness of photovoltaic power generation and EV charging, the dynamic response capability, power support capability, effective convergence time, system stability, system failure rate, and other characteristics of regional loads are comprehensively analyzed, and the grid energy management model of EV charging network and distributed photovoltaic is proposed. Secondly, according to certain statistical characteristics, the distributed photovoltaic will be concentrated, and EV charging will be prioritized to achieve nearby consumption. Finally, different scenarios are described, and the scenarios of charging in the park, community life, and power supply service are selected for analysis. This mode is intended to guide the consumption of new energy through economic leverage, which can realize the unified regulation of distributed energy convergence, consumption and storage.

**Keywords:** photovoltaic output model; electric vehicle consumption (EV-consumption); charging load characteristics; energy management; electricity consumption model

## 1. Introduction

Photovoltaic is the most widely used clean energy. How to improve the efficiency of photovoltaics is an important task on a global scale. Facing the dual-carbon goal of "carbon peaking and carbon neutrality", the Chinese government has proposed a "county-wide photovoltaic" and a new power system construction plan. At present, the forms of photovoltaic power plants can be roughly divided into two types: (1) centralized photovoltaics, and (2) distributed photovoltaics. There is no doubt that the energy Internet is an important carrier for accessing photovoltaic energy, and the key facility is the energy router. In engineering practice, centralized photovoltaics are usually connected through the power transmission network, while distributed photovoltaics are generally connected through the distribution network. In theory, the use of energy router technology can seamlessly absorb photovoltaic energy of various magnitudes, but in practical projects, the main problem is the cost, including the impact of the randomness of photovoltaics on the large power grid. Therefore, how to deploy energy routers and realize the nearby consumption of distributed photovoltaics is a highly feasible and challenging task. At present, the research on energy

routers mainly focuses on two ways: (1) Based on the conceptual model of energy hub, the energy input and output characteristics and coupling relationship of energy routers are modeled [1,2], and the model is applied to planning and optimization of multi-energy systems [3,4], (2) Research on power conversion modules based on power electronic conversion technology [5], or energy management and operation control strategies based on energy routers with deterministic topology structures [5,6].

On the other hand, with the large-scale use of EVs, the charging load mainly depends on the power grid, which has a profound impact on the planning and operation of the power system and the operation of the power market. The charging load has complex characteristics [6–8]. As far as a single vehicle is concerned, it is mainly determined by the user's travel demand and is also affected by the user's usage habits, equipment characteristics, and other factors [9–12]. As far as the regional power system is concerned, it is also affected by the number and scale of EVs and the perfection of charging facilities [13–15]. Due to the uncertainty and mutual difference of user demand and user behavior, the charging load has certain randomness and dispersion [16]. At the same time, the energy storage capacity of the battery makes the user have certain flexibility in the selection of charging time, and the charging load has a certain controllability [17]. Research shows that proper charging control can not only suppress and eliminate the adverse impact of EVs on the power grid, but also promote the operation of the power grid [18], and the benefits of load dispatching are initially shown. In particular, the introduction of V2G (vehicle-to-grid) technology makes it possible to use the energy storage resources of EVs in an average of 96% of the idle time, adjust the charging and discharging process, promote the absorption of renewable energy, and provide auxiliary services for the power grid.

The overall cost of using a distribution network to access distributed photovoltaic is high, which limits the application mode and scope. According to statistics, after a large number of low-voltage distributed photovoltaic systems are connected to the grid, it has a great impact on the voltage change in the station area. Some stations have the problem of "high voltage in the daytime and low voltage at night", that is, when the sunshine is good, the power reverse transmission causes the voltage to exceed the limit; During the peak load period at night, some users have low voltage problems. Therefore, there are a variety of distributed photovoltaic nearby consumption modes under exploration. This paper proposes a method of using EV charging to absorb distributed PV nearby, which can not only improve the utilization rate of PV, but also reduce the interference to the distribution network as much as possible. One of them is to centralize the energy generated by distributed photovoltaic. Its advantage is that each photovoltaic node does not need to install an inverter. At the same time, combined with the corresponding energy storage planning, a large-scale virtual power plant or microgrid system can be formed to improve the system stability, improve the power quality, and improve the regulation flexibility.

The main achievement of this paper is to propose an application mode of an EV charging system and distributed photovoltaic power generation local consumption. By discussing the EV charging network and the grid energy management model including distributed photovoltaic, and according to certain statistical characteristics, the distributed photovoltaic is concentrated, and the EV charging is prioritized for nearby consumption. In addition, the random characteristics of EV random load to deal with distributed photovoltaic are analyzed, focusing on the dynamic response ability, power support ability, effective convergence time, system stability, system failure rate, and other characteristics of the load in the area, dispatching the energy flow formed by photovoltaic, inducing the charging of EVs, and making the photovoltaic consumption in office building areas and residential building areas complementary. Two layers of four supporting model frames are proposed.

- Output layer
  (1) Regional distributed photovoltaic output model.
- Absorbing layer
  (2) Power consumption model of users in the area.



(3) Electric vehicle activity model in the area.

(4) Regional load dispatching model, which is intended to guide the consumption of new energy through economic leverage, so as to achieve the unified regulation of distributed energy convergence, consumption, and storage.

This paper is divided into five chapters. The first chapter mainly analyzes the research background, research status, and related research results, points out the impact of distributed photovoltaic power generation on the stable operation of the grid, puts forward requirements for load side demand response, and analyzes the feasibility of local power consumption and electric vehicle charging. In Section 2, according to the randomness of photovoltaic power generation itself caused by external factors, and the randomness of electric vehicle charging caused by behavioral habits and subjective emotions, the charging network and photovoltaic power generation control system in the park are used to establish a local photovoltaic consumption balance method. The third part mainly analyzes the charging characteristics through data characteristics, considering demand forecasting methods and distributed photovoltaic output models. According to the application characteristics of random to random, M-P rule and single loop theorem are adopted to establish a random matrix, which provides methods for photovoltaic local consumption and energy coordinated control management. The fourth part establishes dynamic management and dispatching methods from the grid layer, photovoltaic power generation system, and energy router. Considering different scenarios, it establishes relevant management mechanisms to achieve unified regulation and management of distributed energy convergence, consumption, energy storage, and economic operation with the help of demand response. Section 5 is an example analysis.

## 2. Absorb Photovoltaic Random Load Nearby

### 2.1. The Basic Idea

Photovoltaic power generation is obviously a random power source. EVs as a whole also belong to random load. When two random factors meet, they often produce a fight-with-fire effect. Photovoltaic output is closely related to sunshine and meteorological conditions. At the same time, multi-agent participation is an important feature of the new power system, including power grid, virtual power plant, load aggregator, EV users, residential users, and park management institutions [19]. Through the absorption of photovoltaic power generation by EVs, the charging time is often limited, and the users' charging behavior is guided through the formulation of peak-valley electricity price, real-time electricity price, or auxiliary service price, etc. The whole electricity consumption process will be optimized with the participation of multiple subjects and relying on specific objective function.

### 2.2. Photovoltaic Energy Control System

In distributed photovoltaic construction in parks and residential areas, photovoltaic capacity is calculated according to the allocated area. The total annual solar radiation and the annual sunshine duration are determined. Taking a distributed photovoltaic power station with 1 MWp capacity in a certain area as an example, calculated by the annual utilization hours of 1300 h, it can generate 1000 × 1300 × 0.8 = 1.04 million kWh per year (system efficiency is 80%). In order to improve the photovoltaic energy consumption level, the energy classification granularity will be as small as possible. In this paper, the distribution granularity is divided into 96 sub-sections for 24 h, which is the same as the electricity metering granularity of general smart meters. Similar to the general microgrid system, the photovoltaic energy management and control system includes power generation prediction, load prediction, real-time scheduling, power quality monitoring and analysis, intelligent electricity consumption, and economic operation of the photovoltaic microgrid subsystems. Through the control of the internal distributed power supply, energy storage system, user load and the point to common coupling with the power grid, the safe, stable, and economic operation of photovoltaic microgrid is realized (Figure 1).

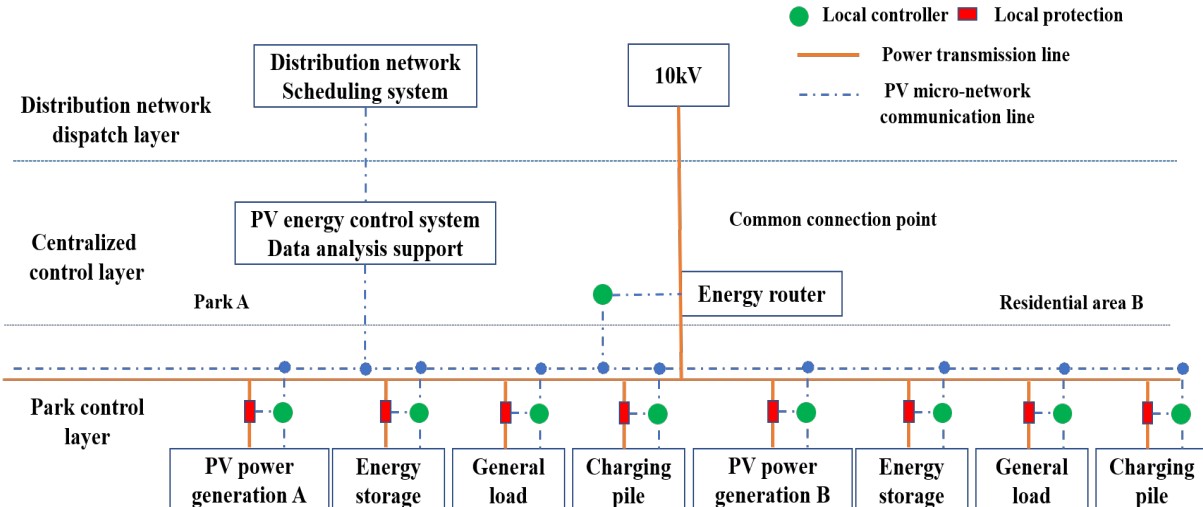

**Figure 1.** Overall block diagram of the system.

(1) **Park control layer**: it mainly involves household, park level distributed photovoltaic power generation unit level self power supply system and energy storage device, as well as demand users including charging piles. When the photovoltaic power generation is sufficient and the power load in the park is low, the supply and demand balance in the park can be achieved.

(2) **Centralized control layer**: the main management unit is the energy router, which has two-way power management and regulation. It is connected to the power grid to achieve metering and energy interaction functions, and manages user loads to achieve energy management. The photovoltaic energy management system realizes the energy management and dispatching of each park under the 10 kV transformer station area and linkage with the distribution grid dispatching.

(3) **Distribution network dispatching layer**: it shares and interacts with the energy management system in real time, realizes the unified management and monitoring of the load and power supply and distribution under the station area, and maintains the timely supply of energy. The solid red line in the figure represents the energy flow, and the broken blue line represents the information communication transmission channel.

### 2.2.1. Generation Power Prediction Subsystem

Photovoltaic power generation and power forecast is based on a high-precision numerical weather forecast, which collects data from distributed power station centralized control and energy management system. Through modeling each power station, the power generation situation of the power station in the future in a period of time can be reasonably predicted. It can predict medium and long term power prediction curves in the next 72 h and 168 h, short-term power prediction curves in the next 24 h, and ultra short-term power prediction curves in the next 4 h and 15 min. The meteorological data is processed and analyzed from the national meteorological Bureau and local meteorological bureau to obtain the high-precision meteorological data used for power prediction to ensure the accuracy of power prediction.

### 2.2.2. Load Forecasting Subsystem

The PV load prediction results provide a basis for reasonably arranging the startup and shutdown of micropower, formulating economic operation strategy, and guiding the charge and discharge management of energy storage.

### 2.2.3. Real-Time Scheduling Subsystem

Real-time scheduling mainly focuses on second-level scheduling control, converting the day-ahead scheduling plan into scheduling commands that can be executed in a short time to maintain the continuous and stable operation of PV. The main functions include grid-connected mode real-time scheduling, solitary mode real-time scheduling, and energy storage real-time scheduling.

### 2.2.4. Power Quality Monitoring and Analysis Subsystem

The power supply inside the photovoltaic microgrid is mainly responsible for energy conversion by power electronic devices. It is necessary to comprehensively monitor the power quality of the photovoltaic microgrid to ensure that the power quality of the photovoltaic system meets the relevant requirements.

### 2.2.5. Intelligent Electricity Subsystem

The intelligent power consumption subsystem collects the data of the internal power consumption load. After collecting the power consumption load, the system gives the guiding power consumption strategy for the internal load of the micro grid according to the real-time power consumption load, and guides users to use electricity according to the guiding strategy.

### 2.2.6. Photovoltaic Microgrid Economic Operation Subsystem

The photovoltaic economic operation subsystem can give different optimal control operation strategies according to different control objectives, considering various factors such as green and clean energy, energy storage and peak-valley electricity price of public power grid, so as to realize the economic operation of photovoltaic system.

## 3. Data Analysis to Support Nearby Consumption Strategies

### 3.1. Demand Forecasting

Single customer clustering model: Extract the power load curve data of a single customer for a whole year $L_i(i = 1, 2, \ldots, 365/366)$ as the input of the K-means clustering algorithm. The power consumption curves of single electricity customers can be clustered into K classes. By analyzing the electricity consumption behavior of a single customer, the law of electricity consumption of a single customer can be more effectively mastered, and the size of the demand prediction basis for a single customer can be used to train the corresponding short-term load prediction model of the K class of a single customer. We selected 180 typical power users in the power supply area in 2019, and conducted cluster analysis on the eigenvectors of their electricity use behavior information. Figure 2a K-means clustering results. Figure 2b–d show the five layer wavelet multi-resolution analysis results of the IV type selected from the (Figure 2a) cluster. It can be found that the selected user has an obvious idle period.

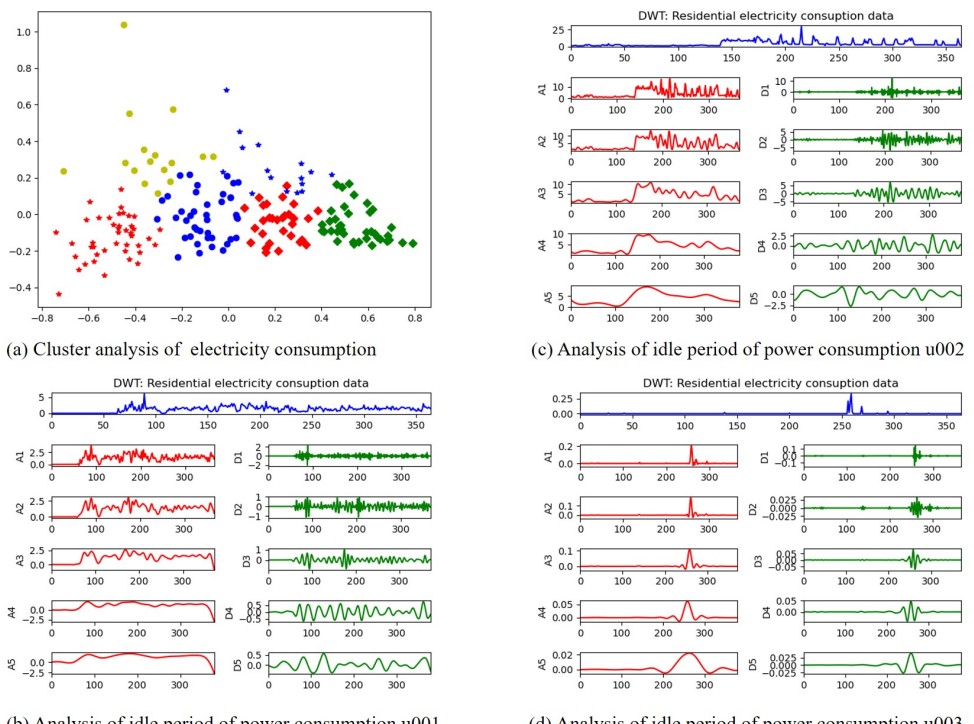

(a) Cluster analysis of electricity consumption

(c) Analysis of idle period of power consumption u002

(b) Analysis of idle period of power consumption u001

(d) Analysis of idle period of power consumption u003

**Figure 2.** Analyzing the electricity consumption behavior with the customer clustering model.

Single customer demand forecast: After analyzing the behavior of a single customer, the prediction model of a single customer is obtained by using a certain kind of daily load curve training model and machine learning algorithm.

In order to predict the short-term load of a single power customer, the prediction step is one or more days in the future, and the load curve of a day is composed of 96, 48, or 24 points, so it is necessary to model the single point. Considering domestic and foreign literature and previous analysis of power customer demand forecasting factors, it is necessary to build a load and influencing factor model at a certain time of day:

$$f_{t_0}(c, d, \vec{P}_n) = P \tag{1}$$

where time $t_0$ is a specific point in the day, $P$ is the actual load value at time $t_0$, $c$ is the temperature value corresponding to time $t_0$, and the load $\vec{P}_n$ of the adjacent day reflecting that the microeconomic status of the power customer is the historical load value corresponding to time $t_0$ of the previous $n$ days. The influencing factor $d$ represents the attribute (working days, holidays, etc.) of the day. After the prediction model at $t_0$ is established, the load at $t_0$ on the forecast day can be predicted by using the influencing factors and the prediction model. Considering that $k$ prediction models are established at $t_0$ according to the historical load data of single power customers, date matching, that is, the characteristics of forecast day and historical day, can be used to match to get the best matching day. Then, the model corresponding to the category of the historical day can be selected to predict the load at $t_0$ on the forecast day.

### 3.2. Distributed Photovoltaic Output Forecast

Weather clustering: Considering that there are many kinds of weather types and the sample size of some weather types is very small, the training samples are insufficient and the modeling is difficult. Based on this situation, this paper takes the actual output data of photovoltaic power generation system and corresponding meteorological information as reference to analyze the meteorological factors affecting photovoltaic output. Based on the historical photovoltaic output data, the expected maximum (EM) clustering method

is used to divide the weather types, and then the training samples are selected from the best clustering set, which can better reflect the actual weather attributes of the day to be measured, and can effectively improve the prediction accuracy of the model. Firstly, all the historical output data of PV in the province are selected to calculate the average output curve data under different weather types. Secondly, the average output curve under the weather type is selected as the feature vector, and the EM clustering method is used to cluster the weather types, so as to realize the clustering of large weather types. Attention should be paid to ensure that the sample number of each class is as equal as possible.

Output forecasting model: In the process of constructing the distributed photovoltaic power generation output prediction model, the following distributed photovoltaic power generation output prediction model is adopted according to the influence factors on the photovoltaic system output.

First, the principle of photovoltaic power generation output is converting the energy of sunlight. The photovoltaic output main external and internal characteristics of solar radiation is $I$ and PV panels $h$. By knowing the solar radiation and the internal characteristics of PV panels, you can set up their $P$ function relationship and photovoltaic power, which can be represented as:

$$f(I, H) = P \qquad (2)$$

Secondly, the reality is that solar radiation data $I$ is difficult to obtain, and the internal characteristics of photovoltaic panels data $H$ are also difficult to be known by power grid companies. Therefore, considering the difficulty of obtaining the data of solar radiation and the change of internal characteristics of photovoltaic panels, it is necessary to analyze the equivalent influencing factors to describe the photovoltaic output.

Solar radiation intensity is closely related to the earth's revolution, rotation, and atmosphere. The solar radiation intensity can be reflected by latitude angle $\alpha$, longitude angle $\beta$, and barrier factor $\gamma$. Equivalently, the function relation between internal and external influencing factors and photovoltaic output $P$, Equation (2) can be transformed into:

$$f(\alpha, \beta, \gamma, \mu) = P \qquad (3)$$

Finally, such parameters as latitude angle, longitude angle, barrier factor and conversion efficiency are still difficult to obtain. Furthermore, latitude angle $\alpha$ and longitude angle $\beta$ reflect the date $d$ and time $t$. The barrier factor $\gamma$ reflects weather type $w$ and temperature $c$. The internal characteristics of PV panels $\mu$ can be used as the historical value of PV output $\vec{P_n}$ under the same external weather type. Then, Equation (3) can be transformed into the following functional relationship model between date, time, weather type, temperature, and similar sunrise force sequence and photovoltaic power generation output:

$$f(d, t, w, c, \vec{P_n}) = P \qquad (4)$$

On the basis of establishing the functional relationship model between date, time, weather type, temperature, and similar sunrise force sequence and photovoltaic power generation output, combined with the previous weather clustering results, the equivalent theoretical model of distributed photovoltaic power generation output prediction is given.

$$f(d, t, w, c, \vec{P_n}) = P \Leftrightarrow \forall_{w_0 \in w} \forall_{t_0 \in t} f'_{w_0 t_0}(d, c, \vec{P_n}) = P \qquad (5)$$

The equivalent theoretical model of photovoltaic power generation output considers the wavelet SVM model of the relationship between date, temperature, and similar sunrise force sequence and photovoltaic power generation output under the condition of constant weather type and time.

First, for weather type, groups $w_1, w_2, \ldots, w_k$ and $t$ were modeled, respectively. For any given weather type $w_0 \in w(i = 1, 2, \ldots, k)$ and the specific time $t_0 \in [t_r, t_s]$ (where $t_r$ is sunrise time, and $t_s$ is sunset time), the corresponding data in the same group of elements with the weather type to be measured are selected to form a training data set, based on

which the prediction model of SVM is constructed. Then, Equation (4) can be converted into the following relation:

$$f_{w_0 t_0}{}'(d, c, \vec{P_0}) = P \tag{6}$$

In order to better describe the time mutation characteristics of photovoltaic output, the wavelet transform is carried out on the photovoltaic output of the adjacent day in the same weather type group $w_i$ when establishing the model:

$$\vec{P_n} = \sum_{a > a_0, b} C_{a,b} * \Psi\left(\frac{t-b}{a}\right) + \sum_{a \leq a_0, b} * \Psi\left(\frac{t-b}{a}\right) = \vec{P_n^\dagger} + \vec{P_n^\ddagger} \tag{7}$$

where $C_{a,b} = \frac{1}{\sqrt{a}} \int_{-\infty}^{+\infty} f(t) * \Psi\left(\frac{t-b}{a}\right) dt$, $\Psi(t_0)$ is the base wavelet.

$$f_{w_0 t_0}{}''(d, c, \vec{P_n^\dagger}, \vec{P_n^\ddagger}) \tag{8}$$

For each weather type $w_1, w_2, \ldots, w_k$, the sunrise time is $t_r$, and the sunset time is $t_s$, with each moment being within $t_0$. According to the weather type $w_i$ and temperature data $c$ obtained from the weather forecast, as well as the historical PV output data under the same weather type, the preliminary prediction result of PV output point-by-point on the day to be measured can be obtained after input into Equation (8).

Using SVM to predict the results will produce a certain error because the PV system output fluctuation is bigger, it is a non-stationary random process, and the PV output prediction of historical data sample quality is higher. If the inadequate training sample data after training the SVM model is not entirely stable, predicted data must generally be within the scope of random fluctuations. Based on the above situation, the PV output prediction error correction model based on Markov chain model can be given by using the adjacent output data of the measured day.

Considering that the system needs to be able to supply power to the load in the station under both grid-connected and off-grid conditions, it is required that the energy storage inverter not only has the inverter function, but also has the ability of off-grid operation, and supports the seamless switch between the grid-connected and off-grid modes, otherwise the load in the station will restart or power will fluctuate.

### 3.3. Analysis of EV Load Characteristics

The nature of the EV charging load mainly depends on the following factors:

1. Travel demand: users' mileage, travel time, travel frequency, travel purpose, etc;
2. Usage habits: charging preferences of users. Differences in usage habits will lead to a certain dispersion of charging loads;
3. Battery characteristics: includes the battery capacity, charge and discharge rate, and charge and discharge curve;
4. Charging facility layout: the distribution characteristics of charging facilities will affect the spatial distribution of the system charging load;
5. Number of EVs: Determines the overall size of the charging load.

As private property, information about EVs is protected by law. The travel data of EVs belongs to the personal privacy data of users. Some EV operation platforms have relevant user mileage, travel time, route, and frequency, but they need the authorization and permission of users to use them and are protected by law. In addition, the characteristics of batteries of different manufacturers and models are different, and the batteries of the same model are also greatly different due to the different use time, frequency, and maintenance degree. Therefore, it brings uncertainty to the analysis of the load characteristics of automobile batteries. Therefore, the load characteristic analysis of EVs has both technical difficulty and legal constraints. The battery characteristic research in factor (3) above does not consider the operation of EVs for the time being, but can be analyzed by the grid interaction data (charging times, charging capacity, electricity payment, etc.). At

the same time, as a public resource, the distribution and grid access mode of the charging pile facilities of (4) can provide basic data support for researchers. Based on this, this paper contributes two main works.

### 3.3.1. Charging Characteristic Analysis

EV load curves are recorded through charging facilities, and charging data are collected at a frequency of 96 points per day, so the comprehensive charging load is $365 \times 24 \times 96$ points of data. A high-dimensional random matrix can be used to model the usage characteristics of the charging facility. When the dimension of the matrix is high enough, the eigenvalues of the random matrix can not only describe the system state, but also confuse the user's private information to a large extent.

Denote a high-dimensional matrix $X$, which is an $N \times n$-order matrix. Denote $X = \{x_1, x_2, \ldots, x_n\}$, where $x_1, x_2, \ldots, x_n$ are independent vectors for each state of the system.

We record the sample covariance matrix: let $X_n = \{x_{i,j}\}_{N \times n}$ be a complex random matrix, the matrix elements $X_{i,j}$ are independently and equally distributed, and the expectation and variance satisfy $E(x_{i,j}) = 0, \sigma^2(x_{i,j}) = 1$, defined as:

$$S_n = \frac{1}{n} \sum_{i=1}^{n} x_i x_i^H = \frac{1}{n} X_n X_n^H \tag{9}$$

is the sample covariance matrix. Where $x_i$ represents the $i$-th column of the matrix $X_n$, and $X_n^N$ is the conjugate transpose matrix of $X_n$. The charging and discharging system and photovoltaic access system are dynamic balance systems. The parameter changes are relatively stable and within a reasonable range during orderly operation. The parameter fluctuates greatly when subjected to large disturbances. From the statistical characteristics of the data, the values of the orderly operation state are distributed in a reasonable range, and the data of the disordered operation state are distributed outside the reasonable range in a certain proportion. In this paper, high-dimensional random matrices are used to evaluate the running state of the system. When the row column ratio of the random matrix remains constant and the number of rows and columns tends to infinity, the limit spectral distribution function of the random matrix has many characteristics, such as Marchenko–Pastur law, single ring law, etc. Considering that the single ring theorem can accurately describe the limit spectrum distribution of the random matrix, its calculation results are also convenient for quantitative analysis, and the inner and outer diameters can be defined as the reasonable interval of the eigenvalue. The out of order operation state is then beyond the reasonable interval. So, the charging characteristics adopt to the following laws:

(1) Marchenko–Pastur law: Suppose a random matrix $X$ of order $N \times n$, $n \to \infty$, $N \to \infty$, and $\lim_{n \to \infty} N/n = c, c \in (0, 1]$, the elements in the matrix satisfy the mean value of 0 and the variance $\sigma^2 = 1$. The empirical spectral distribution of the covariance matrix of matrix $X$ converges to the Marchenko–Pastur law according to the density function:

$$f_{mp} = \begin{cases} \frac{1}{2\pi cx} \sqrt{(b-x)(x-a)} & a \leq x \leq b \\ 0 & x < a, x > b \end{cases} \tag{10}$$

where, $a = (1 - \sqrt{c})^2, b = (1 + \sqrt{c})^2$.

(2) Single-ring theorem: Let $X = \{x_{i,j}\}_{N \times T}$ be a non-Hermitian random matrix, the elements in the matrix are independent and equally distributed random variables, and its expectation and variance satisfy $E(x_{i,j}) = 0, \sigma(x_{i,j})^2 = 1$, for $L$ non-Hermitian matrices $X_i(i = 1, 2, \ldots, L)$, which define its matrix product:

$$X_p = \prod_{i=1}^{L} X_{u,i}^* \tag{11}$$

where $X^*_{u,i}$ is the singular value equivalent matrix of $X_i$. After the matrix product is normalized, the standard matrix product is obtained,

$$X^* = \{\hat{x}_{i,j}\}_{N \times N} \tag{12}$$

When each element in the matrix $X$ is independently and equally distributed, the mean and variance satisfy $E(\hat{x}_{i,j}) = 0$, $\sigma^2(\hat{x}_{i,j}) = 1/N$. When $N$ and $T$ tend to infinity, and $c = N/T$ tends to be constant, the spectrum of the matrix product $X^*$ eigenvalues almost certainly converges to the single ring theorem. On the complex plane, the eigenvalues of the matrix product $X^*$ are distributed in a ring with an inner ring diameter of $(1-c)^{L/2}$ and an outer ring radius of 1.

In the figure, we focus on the load of four nodes in the photovoltaic system charging network. When the charging is stable, all the eigenvalues fall near the ring, as shown in Figure 3a,b. When the charging load is gradually increased and the load changes greatly, it can be seen that the eigenvalue distribution becomes gradually closer to the center of the circle, as shown in Figure 2c,d. When the load increases to a certain extent and the system is close to collapse, the distribution of the eigenvalue is closer to the center of the circle and the distribution range is wider. Therefore, the single loop is used to evaluate the system operation status.

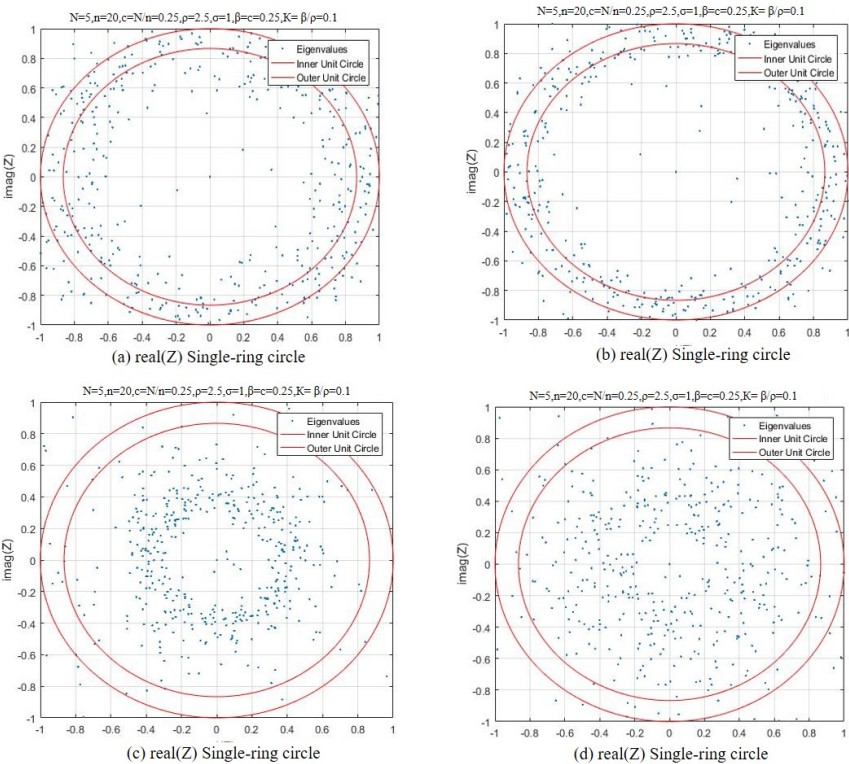

**Figure 3.** Characterizing the charging facility loads. An example of the eigenvalues of a random matrix.

### 3.3.2. EV Travel Preference Analysis

EV mobility information is beyond the control of power grids, virtual power plants, or load aggregators [20]. However, some information can be obtained through the park or community management facilities. For example, video information can be captured from surveillance cameras, structural features of EVs can be extracted, and modeling can be done using federated learning methods. In practical engineering, YOLOv4 framework can be used to achieve the above goals.

The Figure 4 shows the relationship between data processing and local models when forecasting the daily charging probability of EVs, including different data processing methods such as photovoltaic consumption data and user information. This process involves data encryption calculation, information desensitization, and privacy protection processing. MPC (secure multi party computing) in the figure is to solve the problem of collaborative computing between participants such as EV users, charge pile owners, photovoltaic power producers, power grid operators, etc., on the premise of protecting private information and without a trusted third party. In addition, to ensure that user information is not leaked, homomorphic encryption (HE) is used, which can be calculated on the ciphertext without a key. The calculation result is confidential, and needs a key to be decrypted into plaintext. The weight of evidence (WoE) is a method used to measure the difference between the distribution of good samples and bad samples.

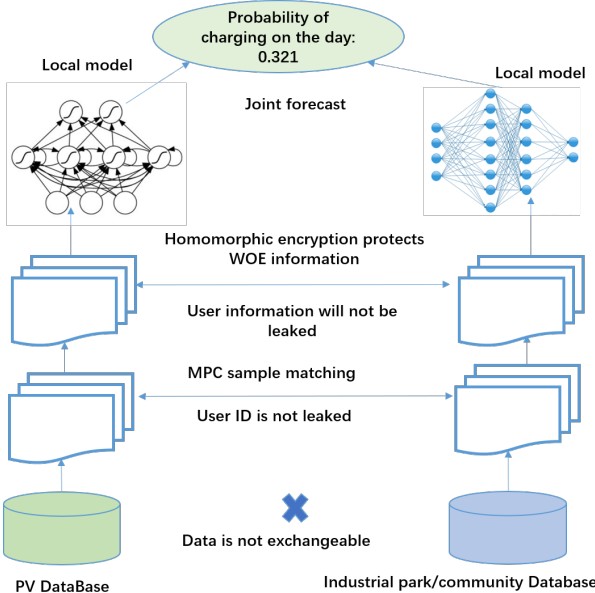

**Figure 4.** Calculating EV charging probabilities with federated learning.

## 4. Operation and Energy Control Strategy

### 4.1. Running the Model

When the characteristics of photovoltaic output, user power consumption, and EV charging are clear, it is necessary to formulate regulation strategies and assist with various economic means. The application scenario is shown in Figure 5. Its power consumption characteristics have different power consumption behaviors during different time periods, and the overall occurrence is random. Relying on the historical power demand data of the power supply area, distributed photovoltaic power generation data, and the requirements for stable operation of the power grid, the power generation user's own needs and charging network load at the terminal side are first guaranteed through smart meters, energy routers, and control equipment from the master station, and the surplus energy is responsible for participating in the power grid dispatching.

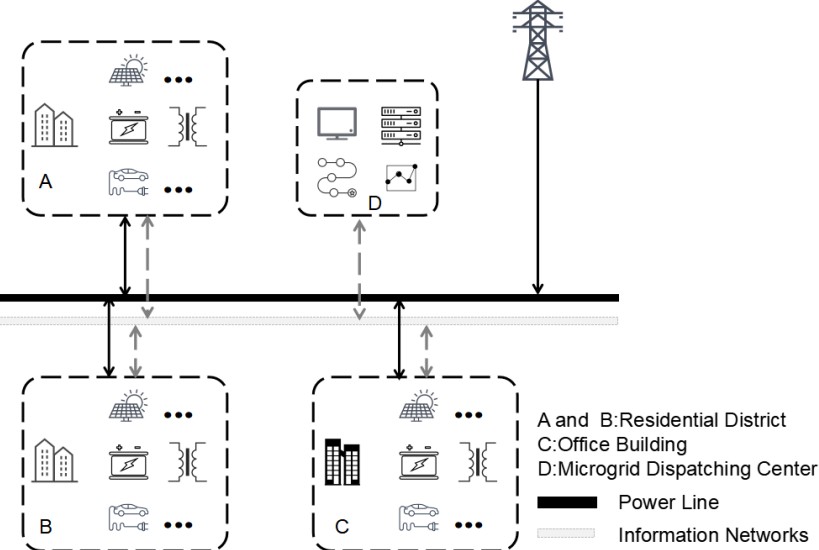

**Figure 5.** Application scenario diagram.

In Figure 5A–C, photovoltaic power generation, automobile charging facilities, power supply management equipment, and power users are integrated. Figure 5D is the micro grid dispatching center, including photovoltaic power generation forecasting, Demand Response (DR) management, and load balance control.

For example, on weekdays, preferential measures are formulated to provide parking spaces and electric car charging services for residential areas adjacent to offices; Rest days are the opposite. The photovoltaic output in residential areas and office areas is preferentially transferred in adjacent areas through energy routers before being connected to the distribution network. The advantage of this is that new energy can be absorbed nearby and the amount of new energy connected to the distribution network can be reduced. Relevant measures can be constrained by a certain objective function. The system runs on the following models (Figure 6):

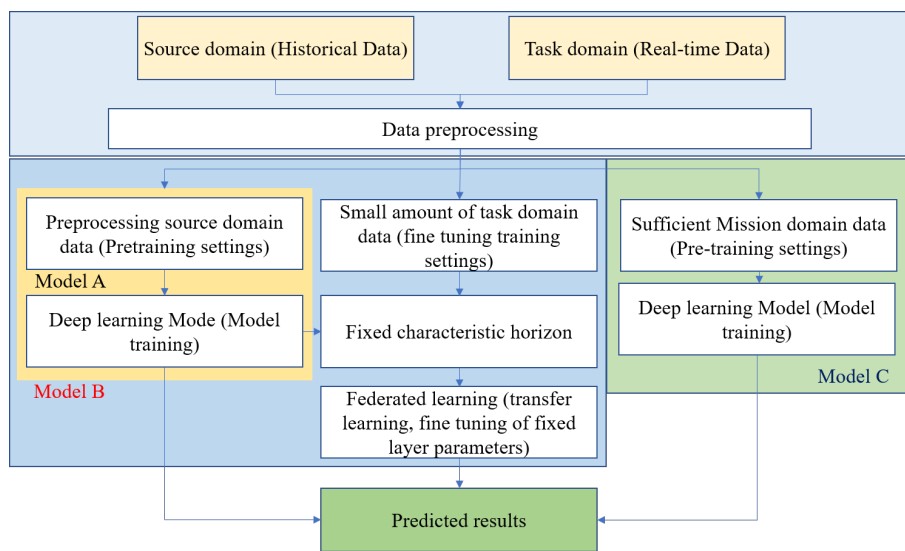

**Figure 6.** Relying on the model training flow chart.

In the regional microgrid under the new power system, the interaction between EVs and distributed new energy to achieve peak regulation and valley filling can be summarized into four typical operation modes, as shown in Figure 7.

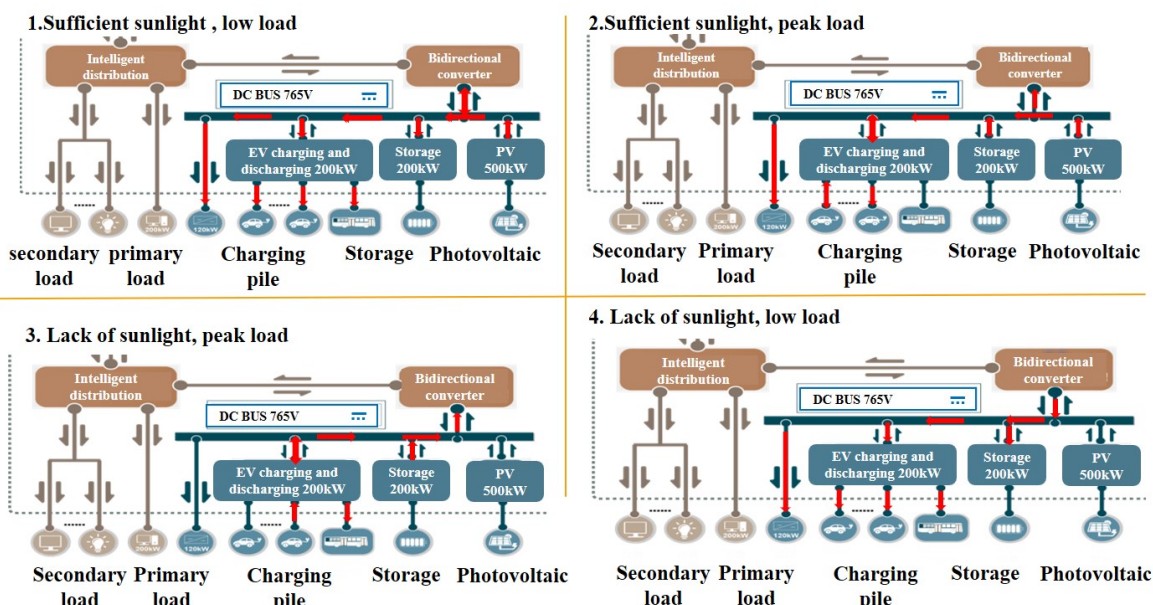

**Figure 7.** Typical operation mode of interaction and energy balance between new energy vehicles and distributed new energy.

The red arrow indicates the power flow direction of energy storage, electric vehicle charging, and photovoltaic power generation. Figure 7 (1) shows that with strong photovoltaic power generation output and low grid load, excess electric energy flows to electric vehicles for charging and energy storage; Figure 7 (2) shows that photovoltaic power generation is prosperous and that the grid load is high. The power flow can solve the electric vehicle charging, the internal energy storage, and output the grid balance load from photovoltaic power generation. Figure 7 (3) shows that energy storage is mainly used to support high load power grid. Figure 7 (4) shows the power supply in the area where the input electric energy at the main grid side solves the light shortage.

(1) **Distributed photovoltaics are sufficient and the load is low.** This scenario is mainly aimed at the urban distribution network from 12:00 noon to 17:00 p.m., when distributed new energy sources supply power to DC loads. At the same time, in order to promote the consumption of new energy on the main network, elastic loads such as energy storage and new energy vehicles are actively using electricity. This enables the distribution network to consume new energy on-site and achieve power balance in the power grid.

(2) **Photovoltaic is sufficient and the load is peaking.** This scenario is mainly aimed at the period from 10:00 a.m. to 12:00 noon in the urban distribution network. When there is sufficient sunlight, the urban electricity consumption is in the morning peak, and the distributed new energy sources are given priority to supply power to the DC load. If the main network has sufficient power, in order to promote the consumption of new energy on the main network, elastic loads such as energy storage and new energy vehicles will actively use electricity. If the main grid is short of power and the power balance pressure is large, the power consumption of new energy vehicles and energy storage equipment will be consumed locally by distributed new energy sources, and no additional burden will be added to the main grid as the goal of operation adjustment.

(3) **Lack of sunlight and peak load.** This scenario is aimed at the period from 18:00 p.m. to 00:00 a.m. in the urban distribution network. During this period, the sunshine gradually weakens until it disappears. The urban electricity consumption is at the evening peak, and the distributed photovoltaics do not generate electricity. We need to guide elastic loads such as energy storage and new energy vehicles to actively participate in grid peak regulation through market-oriented means. The power is transmitted to the AC distribution network through the inverter for the use of the residential users of the

distribution network, reducing the off-grid power of the distribution network from the main network and reducing the burden on the main network.

(4) **Lack of sunlight and low load.** This scenario is mainly aimed at the urban distribution network from 00:00 a.m. to 06:00 a.m. when the urban power consumption is at a low point. Market-oriented measures are adopted to guide elastic loads such as energy storage and EVs to actively use electricity at this time, which not only promotes the consumption of other power at night, but also reduces the dependence of such loads on the power grid during the day.

## 4.2. Energy Dispatch Strategy

Photovoltaic output is a random event, and EV charging is also a random event. During the initial operation of the system, due to the small scale of EVs, photovoltaic energy is absorbed by the need to mount general loads. When there are more EVs, the randomness of the system is enhanced, and the impact of small probability events on the system needs to be estimated. Existing applications generally assume that the randomness of new energy output satisfies the process model, and simulate a series of scenarios through the Monte Carlo simulation method [14]. Charging piles and EVs can be considered to constitute a random service system. For the random service system, the Poisson process is generally used for modeling, and the charging time approximately obeys the normal distribution. Furthermore, the scheduling strategy is actually aimed at the estimated photovoltaic curve to form an EV charging and consumption curve with a co-integration relationship. The purpose is to make the non-stationary photovoltaic output and EV charging form a linear combination, so as to have a stable equilibrium relationship.

Considering that the charging pile and the EV can be considered to constitute a random system, and the photovoltaic output is also dynamically random. In this paper, a cointegration test strategy (EV charging scheduling strategy) is established to analyze the random events of energy scheduling.

**Definition**: Let the EV charging sequence be $X$, and the photovoltaic output be $Y$. Then $X$ and $Y$ obey the relationship of $Y(t) = \alpha + \beta X(t) + \theta(t)$, where the change of the regression coefficient $\beta$ is determined by the charging scheduling strategy. Among them, $\beta$ fluctuates within an interval, restricted by the number of charging piles and EVs.

- When $\beta$ exceeds the upper limit, it indicates that the number of charging piles needs to be expanded; the corresponding strategy is to absorb excess photovoltaic output by the energy storage facility $\theta(t)$.
- When $\beta$ exceeds the lower limit, it means that the photovoltaic output is small. If it is a long-term phenomenon, the photovoltaic scale needs to be expanded.

**Target**: $Y(t) - \beta X(t) \approx c$ (constant)

**Strategy**:

Step 1: Predict the PV output curve (2 h, 4 h, 24 h, 48 h).

Step 2: Schedule EV charging (obtain information on the number of EVs through federated learning).

Step 3: Use the EG two-step test method to test the cointegration relationship.

## 4.3. DR Market Mechanism

Distributed photovoltaics and new energy vehicles have a high degree of dispersion, but the total amount will be larger in the future, and they will both become important users of power grid companies. If grid companies uniformly carry out a market-oriented operation of distributed photovoltaics and EVs, it will not only be difficult and costly, but also the lack of competition will lead to the failure of the market-oriented mechanism [21]. Drawing on the balance settlement unit mechanism of the German power grid, energy aggregation companies are introduced in the operation of new energy vehicles and distributed photovoltaics in the power grid. Each company is equivalent to a DR unit in the power grid. Every EV owner and distributed photovoltaic owner is a customer of the energy aggregation company, and the energy aggregation company integrates the power

generation curve and energy storage capacity of its users. Each energy aggregator will provide its own DR curve to the grid. The introduction of distributed new energy and energy storage equipment will greatly improve the response capability of demand-side response units with EVs as the main body to the power grid. Because new energy vehicles are not complete energy storage equipment, and the cost of energy storage is much higher than the general cost of power generation, in practice, the willingness of new energy vehicle owners to generate electricity must be far less than the willingness to charge, which causes the charging power to be greater than the power generation. The introduction of various energy storage devices will not change the total electricity demand of energy aggregators, but can enhance the DR intensity of energy aggregators, so the curve becomes steeper [22]. EVs need to have the support of big data and power Internet of Things technology to respond to the peak-shaving and valley-filling work of the power grid. At the macro level, EVs are deeply integrated with the grid. The power grid captures the vehicle frequency and charging habits of each EVs owner through big data, and predicts the charging behavior of EVs through big data. The mechanism effectively guides the charging of EVs. In addition, the power grid senses the power of EVs through the Internet of Things technology, predicts the electricity demand of EVs in the future, and predicts the power generation of new energy in advance with the help of weather forecasts, and arranges the power generation plan of the power plant in advance to balance the power of the system. Only by integrating a variety of technical means, starting from the power generation side and the demand side at the same time, and adopting effective market-oriented adjustment methods, can the power balance be achieved in the high-proportion new energy grid. Several aspects involved in demand response are shown in Figure 8.

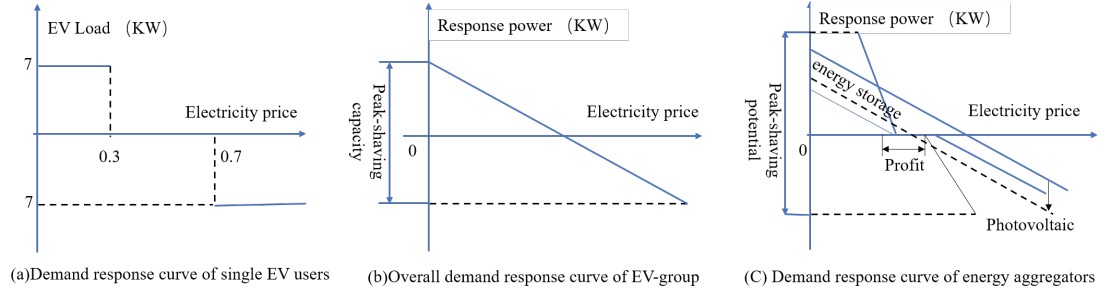

**Figure 8.** EVs and energy storage participate in power DRs.

In the figure, starting from the power generation side and the demand side, only by taking effective market-oriented adjustment measures can the balance of electricity be realized in the high proportion of new energy grid. When charging, EV users set their own expected charging price and expected discharging price. When the electricity price is lower than the expected charging price, the charging pile will automatically charge and charge the vehicle. When the electricity price is higher than the expected charging price, the EV will become the power source and deliver power to the system.

Figure 8a: DR curve of a single automobile user. Suppose a new energy vehicle user uses 7 KW AC charging piles to connect the new energy vehicles to the power grid, and the set expected charge price is 0.3 yuan/kWh, and the expected discharge price is 0.7 yuan/kWh. The user is equivalent to providing his own DR to the power grid.

Figure 8b: Overall DR curve of EV group. When all EV users in the system participate in the DR, the demand curve is superimposed and aggregated. The difference between the upper limit and the lower limit of the curve is the overall peak shaving capacity of the EV group. Because EVs are not complete energy storage equipment, and the cost of energy storage is much higher than the general power generation cost, in practice, the willingness of EV owners to generate electricity is certainly far less than the willingness to charge, which causes the charging power in the curve to be greater than the generating power.

Figure 8c: Introduces energy aggregation companies as DR units in the power grid in the operation of EVs and distributed photovoltaic. The energy aggregation company integrates the power generation curve and energy storage capacity of its users. The introduction of distributed photovoltaic for energy aggregators can enhance the power generation capacity of DR units to a certain extent, so the curve in Figure 8b can be moved down. At the same time, in order to earn reasonable profits, energy aggregators will cause the upper half of the curve to move left and the lower half to move right. The introduction of a variety of energy storage devices will not change the total power demand of energy aggregators, but can enhance the DR strength of energy aggregators, so the curve becomes steeper.

In this mode, the grid company uses the DR curve provided by the energy aggregator to perform the peak shaving work of the grid. When the power grid is at a peak load, it buys electricity from the energy aggregators, and the one with the lower price is preferentially purchased; when the power grid load is at a trough, the power grid sells electricity to the energy aggregator, and the one with the higher price gets it. The key for energy aggregators to earn more profits lies in: improving their own DR capabilities, reducing the cost of energy storage and photovoltaic power generation, optimizing the coordinated operation strategy of photovoltaics, EVs, and energy storage, promoting technological innovation, and providing EVs Users provide cheaper electricity, attracting more new energy vehicle users to join their own DR units. EV users and distributed photovoltaic owners choose between different energy aggregators, and use market competition to choose the most economical energy aggregator to maximize their own interests [23]. For the integrated operation model and energy dispatching mechanism of the charging pile network, photovoltaic power generation and urban distribution network, we analyzed the application effect of four scenarios to achieve the expected goal, as shown in Figure 9.

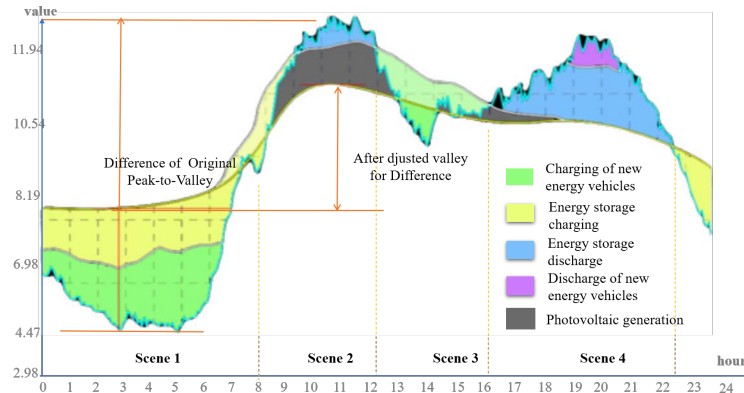

**Figure 9.** Scenario application of A charging station—expectation of an EV participating in DR and operation characteristics. (1) Pure load + energy storage + photovoltaic + new energy vehicles, participate in the scene application. (2) The interaction effect of EVs participating in the power grid: the original peak valley difference is 7.53 MW, and the adjusted peak valley difference is 2.98 MW, reducing the peak valley difference.

## 5. Case Analysis

In this paper, the effectiveness of peak load regulation and valley filling is verified for the EV network and power grid operation strategy. The load group composed of A charging station, B residence, and C residential community is selected as the research object in China, and distributed photovoltaic and energy storage equipment are added to the load group. The power balance in this area is simulated by the Monte Carlo method. A, B, and C are all powered by the No.2 main transformer of an substation. The load curve of the No.2 main transformer of the substation is shown in Figure 10.

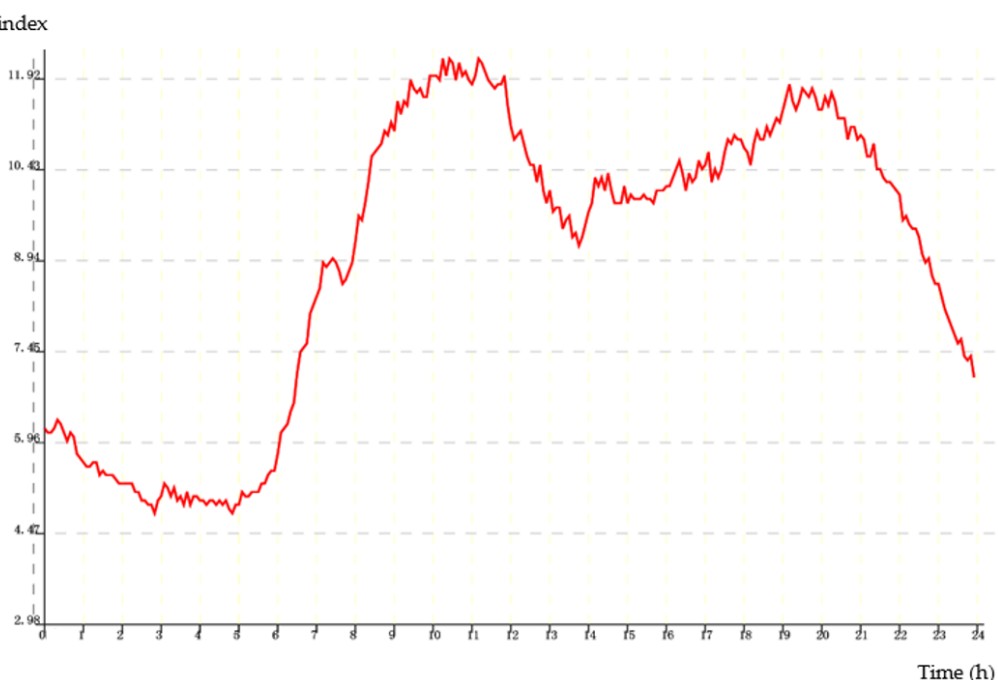

**Figure 10.** DR curves for energy aggregation users.

Without the participation of EVs in peak regulation, the load peak valley difference of the No.2 main transformer of a substation on 8 October 2021 is 5.8 MW. The maximum load occurred at 11:15 noon and the minimum load occurred at 02:50 am. The charging station in block a is equipped with 23 double gun 120 KW DC charging piles and 2 single gun 7 KW AC charging piles, with a maximum load of 2.774 MW. There are 1200 parking spaces in the B mansion and the C residence. If 20% of the parking spaces park new energy vehicles to participate in the peak shaving and valley filling of the power grid in the future, it can provide ±1.68 MW peak shaving resources for the power grid. Distributed photovoltaic, new energy vehicles, and energy storage are used to compensate for the load fluctuation of the power grid. Monte Carlo method is used to simulate the power consumption in this area. As shown in Figure 11, the peak valley difference of No.2 main transformer of an substation can be reduced to 1.72 MW.

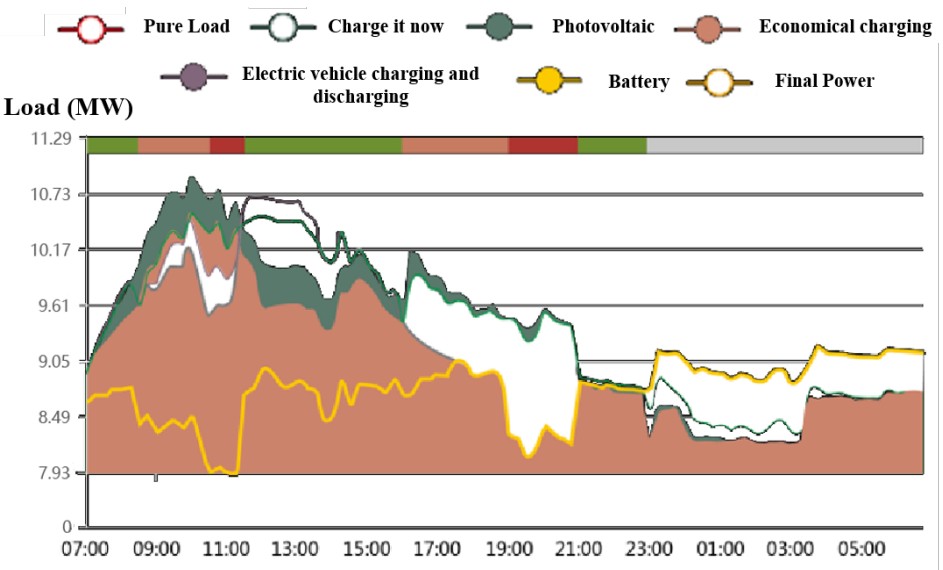

**Figure 11.** Photovoltaic, EVs, and energy storage coordinate peak-shaving and valley-filling.

In order to effectively evaluate the effectiveness of adopting EV charging groups to absorb distributed photovoltaic power generation locally and the coordination of energy conversion, we selected relatively independent distributed photovoltaic power generation, charging pile clusters, and regional A annual data with complete power consumption types for analysis and verification, relying on the park to which the general distribution system belongs. Firstly, according to the M-P theorem and the single loop principle, the stochastic system resource scheduling and energy dynamic balance adjustment involving the stochastic matrix are established. Secondly, according to the user demand response data of the cell, the charging load data in a certain time range, and the photovoltaic output data. As shown in Figure 12, the analysis curve of the smart meters at each node on 15 July 2021 with one 30 min data point shows that the output of smart power devices (charging piles, etc.) and photovoltaic power under the random dynamic mechanism in this paper is effectively balanced and controlled by the external environment and grid energy, which not only effectively absorbs green power from new energy sources, but also provides reference for the stable operation of urban power grids and orderly load management.

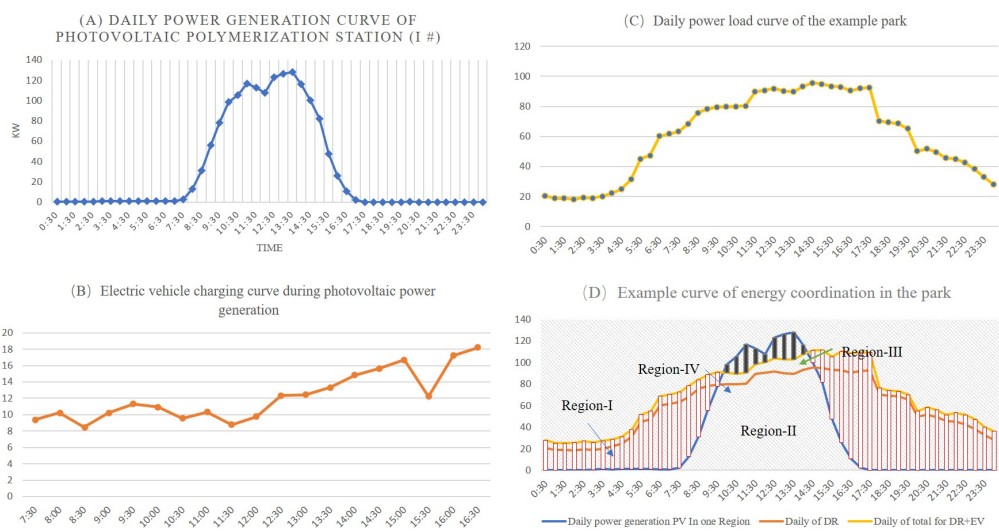

**Figure 12.** Example curve of energy regulation in the park taking into account the on-site absorption of photovoltaic power generation by EVs.

Figure 12A Refers to the daily power generation curve of the distributed photovoltaic polymerization substation in the region I, Figure 12B refers to the charging load during the main output period of the distributed photovoltaic within the scope, Figure 12C refers to the real-time data collection of smart meters for power users, and Figure 12D refers to the energy relationship diagram under the coordinated control of the grid. From Figure 12D, we can see the relationship between the distributed photovoltaic system, power load, and grid power regulation. In the non photovoltaic power generation period, the power supply mainly comes from grid transmission to meet the demand and supply platform, such as in region I; When photovoltaic power generation is in a peak period within a day, such as in region II, it can be used as the main support for regional power demand, or even the demand for excess power generation. Therefore, it can be automatically regulated through the energy router, and the surplus power can be connected to the grid, such as region III. Region IV is mainly used for single day charging.

In order to evaluate the strategies of introducing electric vehicle charging and local consumption of distributed photovoltaic power generation in the community and automobile energy storage to participate in demand response, in Figure 13 we conducted a stability analysis on the operation data of the power supply network in Community C (taking into account the voltage qualification rate, three imbalances, failure rate, harmonic, zero sequence current, etc.). Compared with 2020, the power supply reliability of Community

C in 2021 has significantly improved by about 2%, as shown in Figure 13 (1). In terms of community distributed photovoltaic power generation utilization, the effective power generation utilization data increased by about 8%, as shown in Figure 13 (2), compared with 2020 under the local absorption mechanism of introducing external loads and establishing energy storage strategies. It can be seen that the local photovoltaic power generation consumption under the grid energy management of electric vehicle charging network and distributed photovoltaic has good economy.

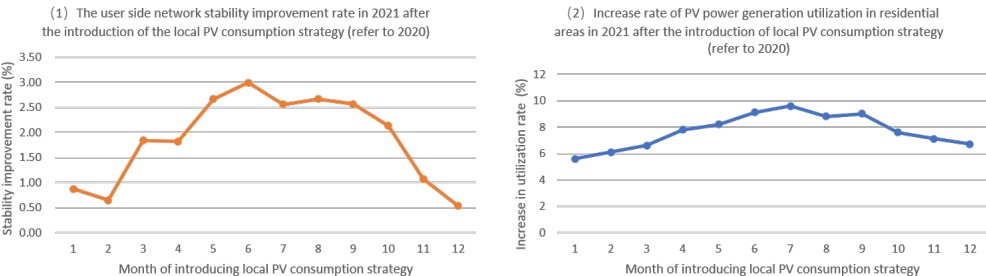

**Figure 13.** Introducing electric vehicle charging and energy storage to participate in the performance analysis and evaluation of the residential distributed photovoltaic power generation absorption strategy (absorption capacity, power supply stability).

## 6. Discussion

The development of smart grids enables end users to actively participate in energy management systems (EMS) through a demand response (DR) strategy. The integration of renewable energy (RES), electric vehicles (EV), and energy storage systems (ESS) provides additional energy and energy storage options for microgrids. Renewable energy generation, market price, electric vehicle drive plan, and other factors determine the efficiency of micro grid operation. Ref. [24] describes the power flow between micro grid components and power grid under random and deterministic methods: (i) the two-way energy trading capacity of an electric vehicle fleet arriving at the office building under a random electric vehicle driving plan, (ii) the impact of photovoltaic uncertainty on an EMS operation based on real intelligent measurement data and compared with deterministic photovoltaic production methods, and (iii) the impact of setting different priority factors on the total system cost when selling energy back to the grid from resources. The goal is to minimize the operating costs involved in customer satisfaction, while taking into account the potential uncertainties, and balance the real-time supply and demand by adjusting the optimal planned charging and discharging of the electric vehicle mobile/local battery storage, grid power supply, and delayed load. Ref. [25] proposed a four stage intelligent optimization control algorithm for the bidirectional charging station of photovoltaic power generation and fixed battery energy storage electric vehicles integrated with the park (community) commercial buildings. Under the goal of effective integration of distributed photovoltaic and electric vehicle charging, there are two situations: one is that electric vehicles can only be charged during working days, and the other is that electric vehicles must be charged seven days a week. An applicable priority mechanism needs to be designed to facilitate charging multiple EVs from a single EV-PV charger. Ref. [26] compared various dynamic charging modes of electric vehicles in order to minimize the dependence on the grid and maximize the use of solar energy for the direct charging of electric vehicles, and solved the problem of charging multiple EVs from a single EV-PV charger.

A key link in the nearby consumption of new energy is short-term power load forecasting. This link can assist the dynamic balance of the power generation end and the power consumption end. With the large-scale grid connection of distributed photovoltaics, the volatility and nonlinearity in the load sequence increase sharply, further increasing the difficulty of forecasting. The accuracy of traditional forecasting methods such as regression analysis, exponential smoothing, and least squares is widely used in engineering. In order

to reduce the influence of volatility and nonlinearity, the combined prediction method based on signal decomposition is often used in the current research; that is, the signal decomposition algorithm is used to decompose the complex power load sequence into simple sub-sequences. Secondly, a corresponding prediction model is established for each subsequence. Classical signal decomposition algorithms, including wavelet decomposition, empirical mode decomposition, and variational mode decomposition, are widely used in the field of power load forecasting.

Based on information security, privacy protection, and cross-data domain information interaction and processing, federated learning is an emerging technology used to solve the problem of data silos. It was first proposed by the Google team in 2016. Federated learning defines a new distributed machine learning framework. Under this framework, virtual models are designed to solve the problem of collaboration between different data owners without exchanging data, which can effectively help multiple institutions meet user privacy requirements, data usage, and machine learning modeling as required by protection, data security, and government regulations. The introduction of federated learning into the power system can effectively solve the problem of existing data islands.

## 7. Conclusions

In order to achieve the 2030 and 2060 goals of "carbon peak and carbon neutrality", relevant Chinese departments have formulated the "whole county photovoltaic" and new power system construction plans. A large scale of distributed photovoltaic constructions and operations will pose challenges to the regional distribution network, energy management, and terminal requirements. The energy router can seamlessly manage the consumption of photovoltaic power generation of different levels. However, due to the large number and wide range of distributed photovoltaic points, the cost of its management and the impact of the randomness of photovoltaic on small power grids are urgent problems that need to be solved. This paper proposes a method of using EV charging to absorb distributed photovoltaics nearby, which can not only improve the utilization rate of photovoltaics, but also minimize the interference to the distribution network, solve the problem of large-scale EV charging loads being too dependent on the power grid, and provide adjustable factors for the planning and operation of the power system and the operation of the power market. In this paper, the model of EV charging and discharging participating in load regulation is established, which provides a good method for DR and energy supply and demand balance, and helps to improve system stability, improve power quality and improve regulation flexibility. As photovoltaic power generation is greatly affected by the weather and other external environment, its output has strong dynamic randomness and belongs to random power supply. The operation of EVs is in essence more affected by external factors and also belongs to random load in general. In this paper, a stochastic mathematical model of "fighting fire with fire" is established. Through the analysis of small probability events, the energy dispatching operation, the charging and discharging characteristics of EVs. and the prediction method of photovoltaic output are analyzed, and the feasibility and economy of local consumption of photovoltaic power generation are discussed. Through the predicted photovoltaic curve, a charge consumption curve of EV with a cointegration relationship is formed. The objective is to form a linear combination of non-stationary photovoltaic output and EV charging, so as to have a stable equilibrium relationship. The photovoltaic power generation is absorbed by EVs, and the charging behavior of users is guided by formulating peak valley electricity price, real-time electricity price, or auxiliary service price. The whole power consumption process will be optimized with the participation of multiple entities and with the help of specific objective functions. In this paper, the living load group composed of charging station A, residence B, and residential community C is taken as the research object. At the same time, distributed photovoltaic and energy storage equipment are added to the load group, which verifies the effectiveness of the operation strategy for peak load regulation and valley filling of the power grid.

**Author Contributions:** All the authors contributions to various degrees to ensure the quality of this work. Conceptualization, methodology, supervision, writing—original draft preparation, formal analysis, Q.H., M.W. and J.G.; software, Z.Z.; validation, L.J.; funding acquisition, investigation, data curation, resources, L.C., P.S. and Z.Z. All authors have read and agreed to the published version of the manuscript.

**Funding:** This research was funded by Science and Technology Project of SGCC OF FUNDER grant number 400-202233168A-1-1-ZN.

**Acknowledgments:** In the process of this research, with the great help of the BCNL research team of Beijing University of Posts and Telecommunications, the 2020 Beijing Laboratory provided an experimental environment for "Research on Advanced Information Networks with Flexible Architecture, Energy-saving and Reliability", and provided effective support for system model simulation. The project was also funded by the "Beijing Co-construction Project Special". In the process of compiling materials, we have received a lot of support and help from the State Grid Gansu Electric Power Company and Gansu Tongxing Intelligent Technology Development Co., Ltd. During the COVID-19 pandemic. I would like to thank my family for their company and care. Finally, I would like to express my gratitude to the individuals and institutions that directly and indirectly supported this research.

**Conflicts of Interest:** The authors declare no conflict of interest.

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
