# Peer review of "Research on a Charging Mechanism of Electric Vehicles for Photovoltaic Nearby Consumption Strategy"

_electronics, doi:10.3390/electronics11203407_

Round 1
Reviewer 1 Report
1. The objectives in the paper are not clear. Please add contributions clearly in the abstract and introduction section.
2. The detailed analysis of load forecasting, power quality monitoring, real time scheduling is missing in the paper.
3. Improve quality of figures. Text is too small.
4. How energy balance has been achieved between distributed resources, loads and EV charger?
Author Response
Dear reviewer
Thank the manuscript reviewer for carefully reviewing our manuscript. You gave us very pertinent and valuable suggestions. We have made an in-depth analysis and correction of relevant problems. Maybe there are some areas that are not well understood. Please correct them and thank you again. Please know that we have improved your question.
Question 1:The objectives in the paper are not clear. Please add contributions clearly in the abstract and introduction section.
Reply 1:Thank you for your analysis and questions!
First,we have made major modifications and adjustments to the abstract, mainly in the following aspects: 1) introduction of research background and problems; 2) Main research results (methods, schemes, etc.), 3) main ideas and solutions. See page1 L1-18.
Second,the content contributed by the article is supplemented (page2-page3: L75-92).
Third,the introduction of main chapters is added (page3 L93-111) .
Question 2 The detailed analysis of load forecasting, power quality monitoring, real time scheduling is missing in the paper.
Reply 2:Thank you for your analysis and questions!
This paper mainly discusses the local consumption of distributed generation, and finds a dynamic energy balance operation mode among the grid, roof distributed generation and user demand. In this paper, there is no detailed research on power generation forecasting, power quality monitoring, real time scheduling and other issues, and we will analyze and study them in other papers later. Thank you for your understanding.
Question 3,Improve quality of figures. Text is too small.
Reply3,thank you very much for your question! We have modified each diagram accordingly.
Question 4,How energy balance has been achieved between distributed resources, loads and EV charger?
Thank you very much for your question. The article adds relevant descriptions (Page11, L363-368). We are still studying the specific realization process of energy balance, and will sort out or publish the relevant results in the future. thank you!

Reviewer 2 Report
This paper provided a study of EV charging mechanism upon the appearance of photovoltaic nearby.
1. Figures: There are not any explanations for Figure 1, Figure 2, Figure 3, Figure 4, and Figure 8. More explanation are required.
2. Page 2: The authors are suggested to summarize the photovoltaic energy control system and its subsystems in a block diagram. The diagram provides a concise illustration.
3. Page 5 Line 200: ‘Equation (1) can be transformed into:’ Do you mean equation (2)?
4. Page 7 Line 254-255: ‘As private property, information about electric cars is protected by law. Therefore, load characteristic analysis of electric vehicles has both technical difficulty and legal constraints. Based on this, this paper has done two main work.’ The relevant factors list are ‘3. Battery characteristics: includes the battery capacity, charge and discharge rate, and charge and discharge curve’ These factors can be obtained. Please specify the technical difficulty and legal constraints.
5. Page 7 Line 270: Please explain why the charging characteristics have to meet the Marchenko-Pastur law and the Single-ring theorem.
6. Page 10 Figure 7: In figure 7-2 and 7-3, the upper level of bidirectional arrows has only one direction. Please cross check.
7. Page 10 (3)Lack of light and peak load: During this period of time, consumers are very likely to travel back to home which makes them difficult to feed back power to grid. Have the authors considered this?
8. Page 10 (4)Lack of light and low load: the introduce of wind power is confusing.
9. Page 13 Figure 9: the color of the figure should be the same with that of the legend.
10. Page 13 Case study: more results should be provided about the charging of EVs as suggested by the title of the paper.
11. The authors are suggested to improve the English. Some of the corrections required are list below.
a. Abbreviations should be introduced before use, for instance EM, PV. Please check all others where applicable.
b. Page 3 Line 107: ‘predict medium - and long-term power prediction curves in the next 168 hours and 72 hours, short’ It is suggested to update it to ‘predict medium - and long-term power prediction curves in the next 72 hours and 168 hours, short’ so that the statement medium and long corresponds to the numbers.
c. Page 3 Line 120: pv should be PV. Please check all others where applicable.
d. Page 5 Line 158-161: ‘customer, is the historical load value of maximum day prior to maximum day corresponding to Maximum t0 moment’ and ‘whether the day is a holiday, the type of holiday, whether it is a weekend, and whether it is a working day.’
e. Page 5 Line 171- 172: ‘this topic with photovoltaic system actual output data and the corresponding meteorological information for reference, analyzes the influence of photovoltaic power meteorological factors,’
Author Response
Thank the manuscript reviewer for carefully reviewing our manuscript. You gave us very pertinent and valuable suggestions. We have made an in-depth analysis and correction of relevant problems. Maybe there are some areas that are not well understood. Please correct them and thank you again. Please know that we have improved your question.
Q1:Figures: There are not any explanations for Figure 1, Figure 2, Figure 3, Figure 4, and Figure 8. More explanation are required.
R1:We have completed the modification(Figure 1, Figure 2, Figure 3, Figure 4, and Figure 8)
Q2:Page 2: The authors are suggested to summarize the photovoltaic energy control system and its subsystems in a block diagram. The diagram provides a concise illustration.
R2:Thank for this question, We have completed the modification (page4, Figure 1).
Q3: Page 5 Line 200: ‘Equation (1) can be transformed into:’ Do you mean equation (2)?
R3:Thank for this question. We have completed the modification.(page7 line237)
Q4: Page 7 Line 254-255: ‘As private property, information about electric cars is protected by law. Therefore, load characteristic analysis of electric vehicles has both technical difficulty and legal constraints. Based on this, this paper has done two main work.’ The relevant factors list are ‘3. Battery characteristics: includes the battery capacity, charge and discharge rate, and charge and discharge curve’ These factors can be obtained. Please specify the technical difficulty and legal constraints.
R4:Thank you for your careful review of the manuscript .We have added relevant content, see page8 L291-303
Q5: Page 7 Line 270: Please explain why the charging characteristics have to meet the Marchenko-Pastur law and the Single-ring theorem.
R5:Thanks to the reviewer for this question. In order to explains ,we added some words as follow: Page7-8,L318-L332
Q6: Page 10 Figure 7: In figure 7-2 and 7-3, the upper level of bidirectional arrows has only one direction. Please cross check.
R6:Thank you for your careful review of the manuscript. We refined the image and annotated it as required. The red arrow indicates the power flow direction of energy storage, electric vehicle charging and photovoltaic power generation. There is no change in the general arrow.
Q7: Page 10 (3)Lack of light and peak load: During this period of time, consumers are very likely to travel back to home which makes them difficult to feed back power to grid. Have the authors considered this?
R7:Thanks to the reviewer for this question, this paper does not specifically consider the contribution to the power grid caused by the increase of consumption home load. Power grid, user load, photovoltaic power generation, etc. are all relatively random. In the article, the impact of relevant random events on energy supply and consumption is considered as a whole.
Q8: Page 10 (4)Lack of light and low load: the introduce of wind power is confusing.
R8Thank the reviewer for this question. This paper mainly considers photovoltaic power generation in residential areas, and wind power is not considered for the time being. The content related to wind power has been adjusted and deleted.
Q9: Page 13 Figure 9: the color of the figure should be the same with that of the legend.
R9:Thank for this question. We have completed the correspondence and matching of legend and figure colors.
Q10:Page 13 Case study: more results should be provided about the charging of EVs as suggested by the title of the paper.
R10:Thank you for your question!
We made a major supplement to the case, and selected representative load areas such as photovoltaic output, electric vehicle charging, and grid coordination from 2021 operation data for analysis and verification.(page17,L482-496,figure12)
Q11:The authors are suggested to improve the English. Some of the corrections required are list below.
Q11-a Abbreviations should be introduced before use, for instance EM, PV. Please check all others where applicable.
R11-a:Thank you for your question! We made improvements as required.
Q11-b:Page 3 Line 107: ‘predict medium - and long-term power prediction curves in the next 168 hours and 72 hours, short’ It is suggested to update it to ‘predict medium - and long-term power prediction curves in the next 72 hours and 168 hours, short’ so that the statement medium and long corresponds to the numbers.
R11-b:Thank you for your question! We made improvements as required.
Q11-c: Page 3 Line 120: pv should be PV. Please check all others where applicable.
R11-c:Thank you for your question! We made improvements as required.
Q11-d Page 5 Line 158-161: ‘customer, is the historical load value of maximum day prior to maximum day corresponding to Maximum t0 moment’ and ‘whether the day is a holiday, the type of holiday, whether it is a weekend, and whether it is a working day.’
R11-d:Thank you for your question! We made improvements as required.(Page6 Line194-198)
Q11-e: Page 5 Line 171- 172: ‘this topic with photovoltaic system actual output data and the corresponding meteorological information for reference, analyzes the influence of photovoltaic power meteorological factors,’
R11-e:Thank you for your question! We made improvements as required.(Page6 Line208-210).
